# Characteristics of Human Nasal Turbinate Stem Cells under Hypoxic Conditions

**DOI:** 10.3390/cells12192360

**Published:** 2023-09-26

**Authors:** Do Hyun Kim, Sun Hong Kim, Sang Hi Park, Mi Yeon Kwon, Chae-Yoon Lim, Sun Hwa Park, Kihak Gwon, Se Hwan Hwang, Sung Won Kim

**Affiliations:** 1Department of Otolaryngology—Head and Neck Surgery, College of Medicine, The Catholic University of Korea, Seoul 06591, Republic of Korea; dohyuni9292@naver.com (D.H.K.); 3unhong@gmail.com (S.H.K.); 2Institute of Clinical Medicine Research, College of Medicine, Catholic University of Korea, Seoul 06591, Republic of Korea; hiparkhi@cmcnu.or.kr (S.H.P.); kmmmmmy@naver.com (M.Y.K.); comeacross@naver.com (C.-Y.L.); 3Postech-Catholic Biomedical Engineering Institute, College of Medicine, The Catholic University of Korea, Seoul 06591, Republic of Korea; pooh0523@nate.com; 4Department of Physiology and Biomedical Engineering, Mayo Clinic, Rochester, MN 55902, USA; gwon.kihak@mayo.edu

**Keywords:** hypoxia, stem cells, turbinates

## Abstract

This study investigated the influence of hypoxic culture conditions on human nasal inferior turbinate-derived stem cells (hNTSCs), a subtype of mesenchymal stem cells (MSCs). It aimed to discern how hypoxia affected hNTSC characteristics, proliferation, and differentiation potential compared to hNTSCs cultured under normal oxygen levels. After obtaining hNTSCs from five patients, the samples were divided into hypoxic and normoxic groups. The investigation utilized fluorescence-activated cell sorting (FACS) for surface marker analysis, cell counting kit-8 assays for proliferation assessment, and multiplex immunoassays for cytokine secretion study. Differentiation potential—osteogenic, chondrogenic, and adipogenic—was evaluated via histological examination and gene expression analysis. Results indicated that hNTSCs under hypoxic conditions preserved their characteristic MSC phenotype, as confirmed by FACS analysis demonstrating the absence of hematopoietic markers and presence of MSC markers. Proliferation of hNTSCs remained unaffected by hypoxia. Cytokine expression showed similarity between hypoxic and normoxic groups throughout cultivation. Nevertheless, hypoxic conditions reduced the osteogenic and promoted adipogenic differentiation potential, while chondrogenic differentiation was relatively unchanged. These insights contribute to understanding hNTSC behavior in hypoxic environments, advancing the development of protocols for stem cell therapies and tissue engineering.

## 1. Introduction

Mesenchymal stem cells (MSCs) belong to a category of versatile stem cells with the ability to be extracted from a range of tissues. Their unique attributes, including self-regeneration capacity, potential to transform into diverse cell lineages, and their ability to influence the immune response, render them an attractive reservoir of cells for employment in regenerative medicine endeavors. Beyond their direct role in differentiating into specialized cells at the site of tissue restoration, MSCs also demonstrate remarkable paracrine effects. These effects are achieved through the secretion of trophic factors and extracellular vesicles, which play integral roles in shaping the microenvironment, facilitating tissue repair, and instigating the regenerative process [1].

In the realm of clinical utility, there is a notable demand for the large-scale generation of MSCs. Nonetheless, the in vitro expansion of MSCs frequently grapples with obstacles, including cell mortality, cellular senescence, and the attenuation of multipotency [2]. As a result, research endeavors have been directed towards examining the optimal culture conditions for augmenting MSCs quality. These investigations center around the enhancement of factors such as media composition, oxygen tension, and substrate coatings. The aim is to bolster the expansion of MSCs, their potential to differentiate, and their broader effectiveness in the therapeutic realm of regenerative medicine. Notably, a substantial body of in vitro studies has been dedicated to probing the effects of hypoxia on diverse facets of MSCs behavior. This encompasses evaluations of viability, proliferation, migration, and differentiation, coupled with a comprehensive exploration of the underlying molecular mechanisms at play [3,4,5]. Because MSCs are usually expanded in the laboratory under normoxic conditions (21% partial pressure of oxygen), they are subjected to higher oxygen stress compared to their physiological niche [6,7]. Cell production for tissue engineering or stem cell therapy is typically performed in vitro. Hypoxic preconditioning, used to increase the efficacy of these cells, can also enhance the denaturation potential during the production, storage, and administration of therapeutic materials [8]. Additionally, hypoxic conditions can impair the ability of MSCs to migrate and adhere [9]. It has been observed that acute and short-term oxygen depletion may induce cell damage, which could lead to apoptosis [10]. Consequently, disparities frequently arise in experimental outcomes concerning MSCs when transitioning from in vitro to in vivo testing environments. Thus, elucidating the precise impacts of hypoxia on stem cells holds the potential to refine and regulate the realm of stem cell therapeutics.

Derived from the human nasal inferior turbinate, human nasal inferior turbinate-derived stem cells (hNTSCs) offer a conveniently accessible resource obtainable during the treatment of chronic rhinitis, and have emerged as a valuable repository of MSCs, as highlighted across various studies [11,12,13,14,15,16,17,18]. These hNTSCs exhibit a remarkable resilience to their microenvironment, showcasing an elevated potential for proliferation and consistently retaining their distinct characteristics throughout successive cell passages [19,20]. Despite their recognition for their MSC potential, the specific impact of hypoxic conditions on these cells has remained an enigmatic realm yet to be explored. Given the fundamental role of oxygen tension in orchestrating cell cultivation and the subsequent implications this holds for the outcomes of experimental pursuits, the investigation into the implications of hypoxia on hNTSCs stands as an imperative and consequential endeavor.

The primary objective of this inquiry was to assess whether hypoxic conditions bring about modifications in the attributes of hNTSCs. The outcomes of this study furnish significant insights that hold the potential to enrich the refinement of protocols pertaining to tissue engineering and the application of stem cell therapy.

## 2. Materials and Methods

This study was meticulously conducted in unwavering compliance with the directives and stipulations set forth by the Institutional Review Board of the Catholic Medical Center Clinical Research Coordinating Center, specifically under the protocol identifier HC15TISI0022. The research also upheld the principles enshrined within the Declaration of Helsinki and meticulously followed the regulatory framework governing informed consent. The Institutional Review Board of the Catholic Medical Center Clinical Research Coordinating Center conducted an exhaustive review, granting official endorsement to all aspects of the study’s protocols. Preceding surgical procedures, all participating patients were duly apprised of the study’s details and willingly furnished written, direct informed consent to partake in the research. Their consent was grounded in a thorough comprehension of the study’s objectives and a genuine willingness to engage in the investigative process.

### 2.1. Donors and Cell Isolation

The study involved the retrieval of discarded inferior turbinate tissues from a cohort of five patients, all aged at least 20 years, who were undergoing partial turbinectomy. The collected tissue specimens from each patient were segregated into two distinct experimental groupings: a hypoxic culture study group and a normoxic culture control group. Exclusion criteria for patient participation encompassed nasal polyposis, antrachoanal polyps, and congenital immunological conditions.

hNTSCs were isolated from the collected inferior turbinate tissues. The five donors, three of whom were male and two of whom were female, were aged between 24 and 47 years old. The tissue samples were first thoroughly rinsed with antibiotic–antimycotic solution (Gibco, Gaithersburg, MD, USA) three to five times, and then washed twice with phosphate-buffered saline (PBS). Following this, the tissues were cut into small 1 mm^3^ pieces. Using culture dishes coated with CELLstart CTS Attachment Substrate (Gibco) following the manufacturer protocols, the tissue pieces were placed in the culture dishes and covered with a sterilized glass coverslip. Next, StemPro^®^ MSCs SFM XenoFree (Gibco), supplemented with 200 mM L-glutamine (Gibco), was added to the culture dishes. The dishes were placed in an incubator set at 37 °C in a 5% carbon dioxide atmosphere. The culture medium was changed every 2–3 days. Following a 3-week culture period, each glass cover slide was removed, and any floating tissues in the culture medium were washed off. The remaining hNTSCs adhered to the bottom of the culture dish were detached using TrypLE Select 10X (Gibco). For the hypoxic condition, hNTSCs following the third passage were placed in a plastic bag (Mitsubishi Gas Chemical Co., Tokyo, Japan); a capsule containing a deoxidizing agent (Sugiyama-Gen Co., Tokyo, Japan) was opened and placed inside the plastic bag along with an oxygen meter (OXY-1; JIKCO CO., Tokyo, Japan). The bag was promptly sealed using a clip. Importantly, the kits containing the deoxidizing agent and carbon dioxide-forming agent were opened immediately before use to limit their reaction upon contact with air.

Subsequent to diligent monitoring utilizing an oxygen meter, it was noted that a notably low oxygen concentration of approximately 5% was established within the enclosed plastic bag, a phenomenon that transpired within a span of 10 to 20 min (Figure 1). Following this step, the plastic bags, which contained the cells, were positioned within a carbon dioxide incubator and subjected to incubation for the designated duration of the culture period. To facilitate the assessment of cellular morphology, a set of cell culture dishes was employed, and the observations were carried out utilizing a phase-contrast microscope (OLYMPUS, Tokyo, Japan). Beyond morphology, the hNTSCs underwent comprehensive scrutiny to gauge any alterations within their immunophenotypic profile, their rate of proliferation, and their capability for multipotent differentiation, all in response to varying oxygen levels.

### 2.2. Cytokines Assays

hNTSCs were seeded at a density of 2 × 10^4^ cells in 24-well plates and allowed to adhere overnight. Conditioned media acquired from hypoxic and normoxic control conditions on days 0, 7, and 14 were stored at −80 °C. Cytokines assays were performed on all five cell lines and average values were calculated. The medium samples were analyzed using the MILLIPLEX MAP human cytokine/chemokine multiplex immunoassay (Millipore, Billerica, MA, USA) to detect various interleukins (IL-1α, IL-1β, IL-4, IL-6, IL-8, IL-10, IL-12), IP-10 (CXCL10), RANTES (CCL5), tumor necrosis factor (TNF)-α, granulocyte-macrophage colony-stimulating factor (GM-CSF), and interferon (IFN)-γ in accordance with the manufacturer’s instructions. These experiments were conducted independently on at least three occasions using individual MSCs donor pools.

### 2.3. Proliferation Assay

To establish growth curves, hNTSCs were simultaneously seeded into 96-well tissue culture plates at a density of 1 × 10^4^ cells per well. Proliferation assays were conducted on all five cell lines and average values were calculated. Subsequently, the cells were subjected to a hypoxic environment for a period of 36 h. Over the course of the experimental duration, the culture medium was renewed every 2 days. To assess cell proliferation, a cell counting kit (CCK-8; Dojindo Laboratories, Kumamoto, Japan) was employed in accordance with the manufacturer’s guidelines. In a concise outline, the existing culture medium was removed, and a fresh medium containing 10 μL of the CCK-8 reagent was introduced, amounting to a total volume of 100 μL per well. The plates were then placed within an incubator set at 37 °C for a duration of 2 h, during which time the cells underwent assessment for viability spanning 7 days. Cellular proliferation capability showed similar patterns in the cell lines evaluated. The optical density measurements were taken in triplicate, comparing against a reagent blank, with a test wavelength of 450 nm and a reference wavelength of 630 nm.

### 2.4. Characterization by the Analysis of Cell Surface Markers on hNTSCs

Flow cytometry analysis was conducted on one hNTSC cell line to scrutinize its cell surface markers. The cells were introduced into test tubes (BD, Franklin Lakes, NJ, USA) at a concentration of 1 × 10^5^ cells per mL and subsequently underwent triple washing using PBS. In the case of the hypoxic group, the cells were exposed to hypoxic conditions for a span of 14 days through the hypoxic cultivation technique. Following this, the cells were incubated with primary antibodies, which included monoclonal antibodies targeting CD14, CD29, CD34, CD73, CD90, and HLA-DR. These antibodies were applied at concentrations that ensured saturation, and the incubation persisted for a duration of 1 h. All the employed anti-human CD antibodies were procured from BD Biosciences (San Jose, CA, USA).

Subsequent to the incubation period, the cells were subjected to three rounds of buffer washing and subsequently centrifuged at 400× *g* for a duration of 5 min. The ensuing step involved the resuspension of cells in ice-cold PBS, which was promptly followed by an incubation with the secondary antibody. This incubation transpired within a light-free environment, at a temperature of 4 °C, persisting for a duration of 30 min. The execution of flow cytometry was facilitated through the use of a FACS Canto II (BD), and the acquired data were subjected to analysis employing the FACSDiva 8.0.3 software (BD).

### 2.5. Multilinaege Differentiation Potential of hNTSCs

Histology was analyzed for each cell line. Normoxic and hypoxic hNTSCs were seeded in 24-well plates (2 × 10^4^ cells/well) with StemPro^®^ Osteogenesis, Adipogenesis, or Chondrogenesis Differentiation media (Gibco), following the manufacturer’s protocols, for 14 days. The hypoxic cultivation method was applied throughout the differentiation period. Complete medium changes were conducted every 2–3 days. After the 14-day incubation period, the cultures were assessed for differentiation using lineage-specific biologic stains.

The evaluation of differentiation was undertaken through histological staining coupled with microscopic observation. For adipogenic differentiation, the application of Oil Red O stain facilitated the visualization of lipid droplets, serving as markers of adipocyte formation. Chondrogenic differentiation, on the other hand, was evaluated through the utilization of Alcian blue stain, which enabled the visualization of sulphated extracellular matrix structures. As for osteogenesis, the staining process involved the utilization of alkaline phosphatase stain. These techniques collectively facilitated the comprehensive assessment of the diverse differentiation pathways. Adipocyte differences using images of Oil Red O staining were analyzed with ImageJ program (Laboratory for Optical and Computational Instrumentation, Medison, WI, USA).

### 2.6. RNA Extraction and RT-PCR of hNTSCs

The assessment of gene expression corresponding to the distinct differentiation states was executed through real-time polymerase chain reaction (RT-PCR). RNA extraction from cells undergoing chondrogenic, osteogenic, and adipogenic differentiation was achieved utilizing the RNeasy Plus Mini Kit (QIAGEN, Valencia, CA, USA). To ensure the absence of genomic DNA, the samples underwent treatment with deoxyribonuclease 1 (QIAGEN). Subsequently, a total of 125 ng of purified RNA was subjected to reverse transcription, resulting in the generation of first-strand complementary DNA. This process was carried out employing the CellScript cDNA Synthesis Master Mix (CellSafe, Suwon, Republic of Korea), encompassing the incorporation of a step aimed at eliminating genomic DNA (QIAGEN).

RT-PCR amplification and relative quantification was performed using TaqMan gene expression assays (Applied Biosystems, Foster City, CA, USA) for the genes osteocalcin, type I collagen, runt-related transcription factor 2 (Runx2), aggrecan, peroxisome proliferator-activated receptor gamma (PPARγ), and acylCoA synthetase (ACS). These assays enabled the precise measurement of gene expression levels in hNTSCs under different experimental conditions (Table 1). The reactions were conducted on a LightCycler 480 PCR System (Roche, Mannheim, Germany), performed in triplicate in a 10-μL reaction volume using TaqMan Master Mix (Roche). Each reaction used 12.5 ng of complementary DNA. Glyceraldehyde 3-phosphate dehydrogenase (GAPDH) served as an endogenous control. All assays used a similar amplification efficiency, and a delta cycle threshold experimental design was applied for relative quantification. The results were analyzed using LightCycler 480 Software ver. 1.2 (Roche).

### 2.7. Statistical Analysis

Statistical analyses were conducted using R statistical software (v4.3.1., R Foundation for Statistical Computing, Vienna, Austria). The statistical significance of differences between groups was determined using Student’s *t*-test and one-way analysis of variance (ANOVA). A *p*-value < 0.05 was considered to indicate significance.

## 3. Results

### 3.1. Histology and Flow Cytometric Comparison of Hypoxic and Normoxic hNTSC Characteristics

The hNTSCs cultured under hypoxic and normoxic conditions were both negative for hematopoietic markers (CD14, CD34, and HLA-DR) and positive for MSCs markers (CD29, CD73, and CD90) (Figure 2). Thus, both groups exhibited the characteristic phenotype of MSCs, and did not differ significantly.

### 3.2. Cytokine and Chemokine Secretion Patterns of Hypoxic and Normoxic hNTSCs

We measured various cytokines and chemokines involved in immunomodulation, including IL-1α, IL-1β, IL-4, IL-6, IL-8, IL-10, IL-12, IP-10 (CXCL10), RANTES (CCL5), TNF-α, GM-CSF, and IFN-γ. Among them, IL-6, IL-8, IP-10 (CXCL10), GM-CSF, TNF-α, and RANTES (CCL5) were detectable at mean values > 1 pg/mL in the supernatants harvested from both the normoxic and hypoxic hNTSC cultures.

Many cytokines and chemokines showed similar secretion patterns between the hypoxic and normoxic groups. In both culture groups, the levels of GM-CSF, IL-6, IL-8, IP-10 (CXCL10), RANTES, and TNF-α increased, whereas IFN-γ levels decreased, over the cultivation period. Notably, the IL-1α and IL-1β levels differed markedly between the two groups after 2 weeks of culture; however, the levels of these cytokines were around or less than 1 pg/mL, so the differences were not significant (Figure 3). Overall, the results suggest that hypoxic culturing of hNTSCs does not influence their immunological properties.

### 3.3. Proliferation of Hypoxic and Normoxic hNTSCs

Cell proliferation was observed over a 7-day period. During day 1, hNTSCs were in a stationary phase, with limited growth. After day 1 through day 7, the cells exhibited rapid and robust growth, with a significant increase in proliferation rate. The hypoxic culture group exhibited a similar proliferation pattern as the normoxic control (Figure 4). There were no significant differences in proliferation rate between the two groups over the 7-day observation period, although the proliferation rate appeared to be higher in the control group than the hypoxic culture group on days 3 and 7. These results indicate that hNTSCs exhibit consistent proliferation and growth patterns under different oxygen saturation conditions.

### 3.4. Multilineage Differentiation Potential of Hypoxic and Normoxic hNTSCs

Next, we histologically evaluated the adipogenic, osteogenic, and chondrogenic multipotent differentiation potential of expanded hNTSCs grown under hypoxic and normoxic culture conditions (Figure 5A, Figure 6A and Figure 7A). Adipogenic differentiation was confirmed via Oil Red O staining of lipid droplets. Osteogenic differentiation was confirmed via alkaline phosphatase staining of calcium deposits. Chondrogenic differentiation was confirmed via Alcian blue staining of sulphated extracellular matrix.

Additionally, the gene expression of differentiation factors was quantitatively analyzed via RT-PCR. Cells exposed to the differentiation media displayed consistently increasing expression levels of mRNAs encoding type I collagen, Runx2, and osteocalcin (osteogenic); SOX9 and aggrecan (chondrogenic); and PPARγ, C/EBPα, and ACS (adipogenic) (Figure 5B, Figure 6B and Figure 7B).

However, there were significant differences in the differentiation capacities of hNTSCs grown under hypoxic versus normoxic conditions. Type I collagen, Runx2, and ACS were significantly upregulated in the normoxic group compared to the hypoxic group. By contrast, SOX9 and aggrecan were similarly upregulated in both the hypoxic and normoxic groups. PPARγ and C/EBPα were significantly higher in hypoxic condition compared to the normo group These results suggest that hNTSCs cultured under hypoxic conditions have lower osteogenic differentiation potentials, but similar chondrogenic differentiation potential and higher adipogenic differentiation potential compared to hNTSCs cultured under normal oxygen saturation.

## 4. Discussion

Our study revealed that hNTSCs subjected to hypoxic conditions maintained their distinctive MSC phenotype. This assertion was substantiated through FACS, which confirmed the absence of hematopoietic markers and the presence of MSC markers. Hypoxia did not exert any discernible impact on the proliferation of hNTSCs. Furthermore, the cytokine expression patterns exhibited noteworthy similarities between the hypoxic and normoxic cohorts throughout the duration of cultivation. Nonetheless, it is noteworthy that hypoxic conditions did suppress the osteogenic and promote adipogenic differentiation potential of hNTSCs, while chondrogenic differentiation remained relatively unaffected. In this study, a duration of 72 h of exposure to hypoxia, as evaluated through a proliferation assay, did not yield any significant reduction in cellular proliferation. On the other hand, prolonged exposure over a period of two weeks to hypoxic conditions, as assessed through a differentiation assay, resulted in a notable decline in cell density. The precise underlying mechanisms governing the demise of MSCs under conditions of oxygen deprivation remain unknown. Nonetheless, it is noteworthy that a prior study reported analogous outcomes, indicating that MSCs viability did not appear to be compromised during brief periods of hypoxic exposure (less than 72 h); however, exposure extending to 120 h led to an escalation in cell mortality rates [21]. Grayson et al. documented that prolonged cultivation of MSCs under hypoxic circumstances resulted in diminished cellular proliferation, although it did not result in a concomitant elevation in apoptosis rates following 9, 16, or 24 days of culture. Hypoxia, as an explanation, exhibited inhibitory effects on their progression through the G1 phase of the cell cycle. Remarkably, despite the extended lag phase, MSCs exhibited limited variability in their expression of p21 and p53 at the initial stages of hypoxic culture, indicative of the fact that hypoxia did not induce the expression of these proteins associated with the regulation of apoptosis in MSCs [22]. Consequently, these findings, when taken in conjunction with our own observations, postulate that hypoxia primarily induces a modest degree of cell death, while the surviving MSC population remains capable of proliferative activity.

Additionally, it has been documented in the literature that low oxygen tension levels exert suppressive effects on the osteo-plasticity of MSCs [21,23]. These effects manifest as an absence of osteogenic differentiation under conditions of 3% oxygen [23]. Furthermore, exposure of MSCs to hypoxia has been observed to result in a persistent downregulation, extending up to 14 days post-exposure, of critical markers such as cbfa-1/Runx2, osteocalcin, and type I collagen [21]. These collective findings provide a plausible explanation for the observed reduction in osteogenic differentiation capacity in MSCs consequent to prolonged exposure to hypoxic conditions. C/EBPδ is a critical determinant for the early stage of adipocyte differentiation. C/EBPδ are expressed in the early stage of adipogenesis in culture and induce PPARγ and C/EBPα expression [24]. Therefore, it is possible to explain that although the expression levels of C/EBPα and PPARγ were lower than control on day 7, they both significantly increased compared to control on day 14. However, considering that PPARγ and C/EBPα are critical transcription factors in adipogenesis [25], based on the results of PPARγ and C/EBPα, it can be judged hypoxic conditions promote adipogenic differentiation compared to the control group.

MSCs have garnered significant interest as a promising avenue for cell therapy and regenerative medicine. Nonetheless, disparities often emerge in preclinical investigations when comparing outcomes between in vitro and in vivo experiments. One plausible rationale for these variations is the contrasting oxygen concentrations inherent in the two environments. While MSCs experience normoxic conditions (21% oxygen) during in vitro cultivation, the oxygen concentration in vivo ranges only between 1% and 9% [4,26]. Taking into account that the oxygen tension within a potential administration site plays a pivotal role within the stem cell milieu, we undertook prior efforts to quantify the oxygen tension in the inferior turbinate. Through these endeavors, we substantiated the presence of a hypoxic oxygen tension within the native inferior turbinate tissue [27].

Within the confines of laboratory conditions, hypoxia has been documented to trigger alterations in both the transcriptome profile and gene-specific activities of spliceosomes in MSCs [26]. Furthermore, in instances of acute hypoxia, the possibility of MSCs succumbing to cellular demise and therapeutic inefficacy arises [28]. As the period of exposure to hypoxia endures, MSCs display swift acclimatization to the surrounding microenvironment by transitioning their metabolic pathways towards anaerobic glycolysis. Notably, even amid this metabolic shift, MSCs manage to sustain their undifferentiated multipotent state [10].

A multitude of MSCs exhibit sensitivity to variations in the oxygen levels within their immediate surroundings [10]. Oxygen limitation can trigger diverse intracellular mechanisms that orchestrate the development of an adaptive response, including modifications in gene expression. Consequently, data concerning the response of MSCs to hypoxic conditions have yielded somewhat inconsistent outcomes, highlighting both detrimental and beneficial effects [3]. Moreover, the modifications observed in the characteristics of MSCs when subjected to hypoxic conditions have the potential to introduce complexities that impinge upon the reliability of MSC-based stem cell products. These products have undergone prior validation within tightly controlled in vitro contexts, adding an additional layer of complexity to their real-world applicability.

Initially, our speculation was inclined towards the notion that a hypoxic culture condition might confer a stimulatory effect on hNTSC proliferation. However, the reduction in oxygen consumption due to heightened carbon dioxide levels elicited an adverse impact on their proliferative capacity [29]. This antagonistic association could be attributed to multiple factors. To begin with, mitochondrial function holds a pivotal role in governing cell proliferation by generating cellular energy in the form of ATP. Elevated carbon dioxide levels interfere with mitochondrial function through the induction of miR-183 and the subsequent downregulation of IDH2, culminating in a hindrance to cell proliferation regardless of the hypoxic milieu [29]. Moreover, the impairment of stem cell capacity is a collateral consequence of carbon dioxide-induced mitochondrial dysfunction [30]. In certain instances, efforts to mimic hypoxic conditions indirectly have entailed the induction of hypercapnic conditions. However, this approach may not holistically replicate the ideal microenvironment inherent in the stem cell niches within the submucosal region of the nasal turbinate. Additionally, hypercapnic conditions can exert influences on cellular viability and metabolic pathways. In contrast, our study undertook a distinct approach by directly inducing low oxygen tension through the application of a deoxidizing agent, facilitating the absorption of oxygen. This intervention was concomitantly accompanied by continuous monitoring of oxygen levels through a dedicated meter. The culmination of these efforts translated to the creation of an authentic direct hypoxic culture environment, within which the process of in vitro expansion unfolded.

By employing direct hypoxic cultivation techniques, we were able to ascertain that exposure to hypoxic conditions held no influence over the proliferation or the secretion of cytokines and chemokines by hNTSCs. These findings provide compelling evidence that hNTSCs exhibit a notable resilience against the constraints of oxygen deprivation. In preceding studies, we corroborated these comparable outcomes across both in vitro and in vivo examinations of hNTSCs, encompassing their neurogenic, osteogenic, and chondrogenic differentiation capacities, as well as their presence in the airway mucosa [11,12]. However, it is noteworthy that hypoxic conditions did attenuate the osteogenic and promote adipogenic differentiation potential of hNTSCs, while chondrogenic differentiation remained relatively unaffected (although not statistically significant, differentiation ability decreased in PCR analysis). Additional research will be needed to confirm differentiation through additional tests such as Western blot or immunofluorescence. Building upon the insights derived from the current study, it stands to reason that the inherent robustness of hNTSCs when faced with hypoxic conditions could indeed be a substantial contributor to the coherence observed in our earlier investigations, conducted across in vivo and in vitro platforms. Nonetheless, to substantiate this hypothesis, it is indispensable to undertake further comparative experiments.

## 5. Conclusions

Within the scope of the ongoing study, we definitively validated the preservation of hNTSC characteristics under hypoxic culture conditions, showcasing a similarity to their in vivo counterparts, as opposed to the normoxic culture conditions. These outcomes strongly indicate that hNTSCs possess the capability to uphold their MSC-like attributes, thereby positioning them as a promising candidate for potential application in stem cell-based therapeutics. This revelation holds the potential to expedite the formulation of meticulously designed and efficacious protocols that cater to the domains of tissue engineering and stem cell therapy, fostering a more streamlined and successful trajectory in these areas of research and practice.

## Figures and Tables

**Figure 1 cells-12-02360-f001:**
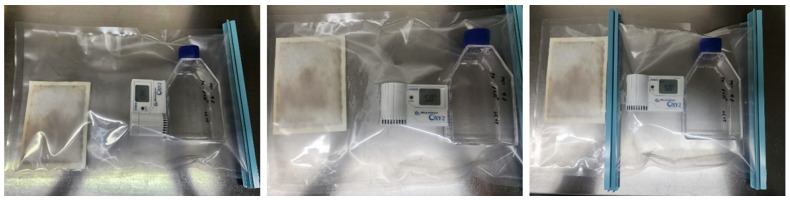
Schematic of the hypoxic cultivation set up. hNTSCs and a gas-controlling agent were placed in a gas-barrier bag, which was sealed with a clip to lower the oxygen saturation within the bag. The oxygen meter showed an initial oxygen level of 13.4% (**left**). After 10~20 min, the oxygen meter showed that the desired oxygen level of 5% was achieved (**middle**). Once reaching 5% oxygen, the gas-controlling agent was isolated from the hypoxic cell cultures using another clip (**right**).

**Figure 2 cells-12-02360-f002:**
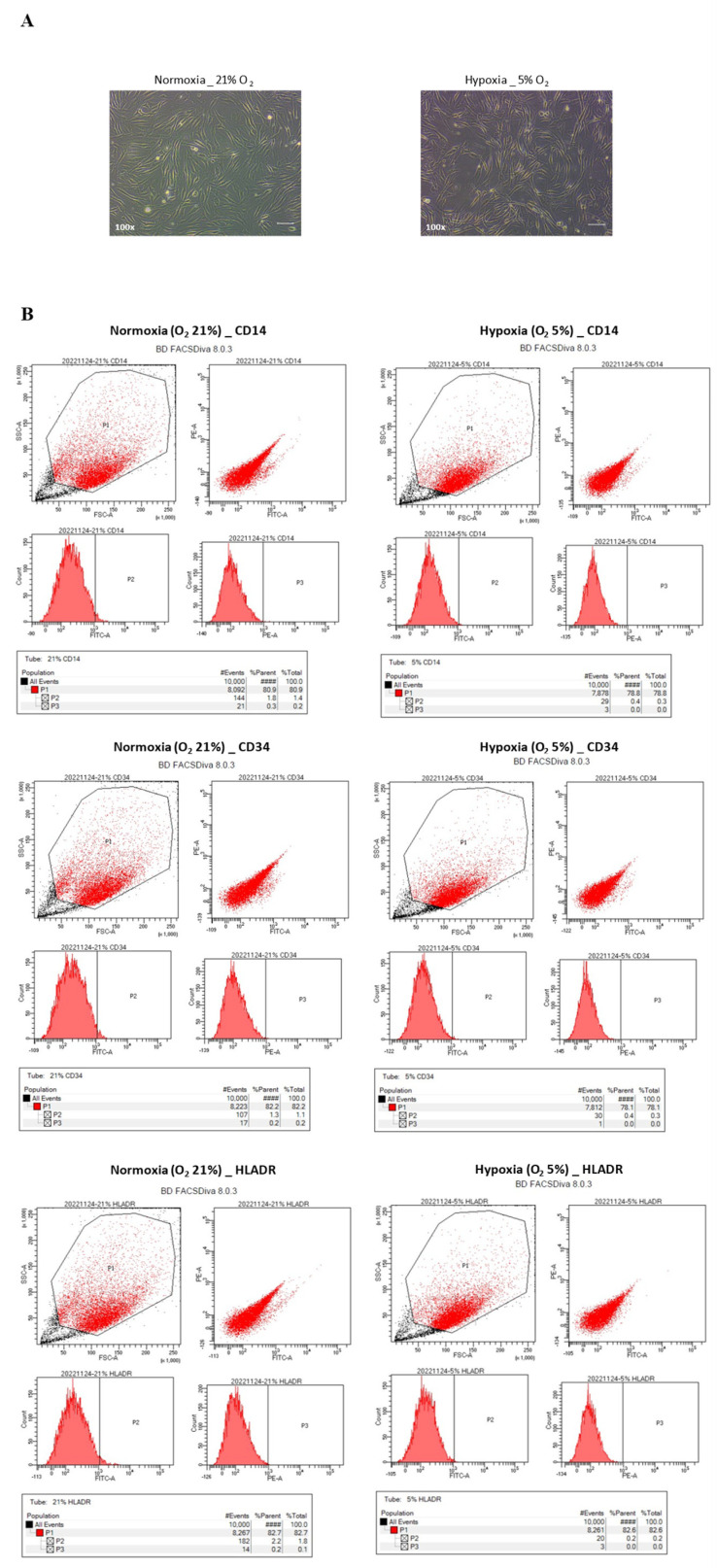
Cell morphology after primary explant culture (**A**) and fluorescence-activated cell sorting analysis (**B**) of hNTSCs cultured under hypoxic and normoxic conditions (sample, *n* = 1). Cells in both groups adhered to the culture dish and displayed a similar spindle-shaped, fibroblast-like morphology (100× magnification). Scale bar: 10 μm (**A**). Flow cytometry analysis after three passages confirmed that hNTSCs from both groups were positive for CD29, CD73, and CD90, and negative for CD14, CD34, and HLA-DR (**B**).

**Figure 3 cells-12-02360-f003:**
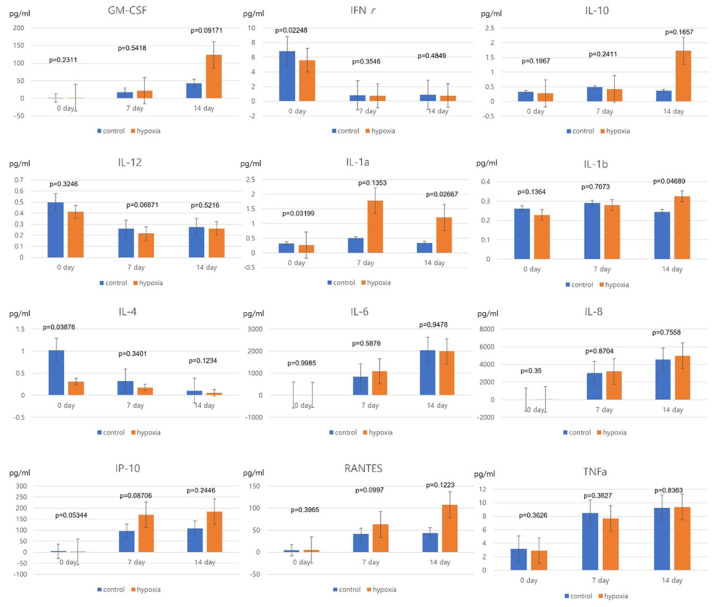
Effects of serum-free cultivation on cytokine and chemokine secretion by hNTSCs (sample, *n* = 5). The supernatants of hNTSC cultures under normoxic and hypoxic conditions were evaluated via enzyme-linked immunosorbent assay for the secretion of cytokines and chemokines IL-1α, IL-1β, IL-4, IL-6, IL-8, IL-10, IL-12, IP-10 (CXCL10), RANTES (CCL5), TNF-α, GM-CSF, and IFN-γ. In both groups, the levels of GM-CSF, IL-6, IL-8, IP-10 (CXCL10), RANTES, and TNF-α increased during the cultivation period, whereas the IFN-γ level decreased during the cultivation period. These cytokine and chemokine secretion patterns were similar to those of hNTSCs from normoxic cultivation. Error bars are standard errors. *t*-test was used for statistical analysis.

**Figure 4 cells-12-02360-f004:**
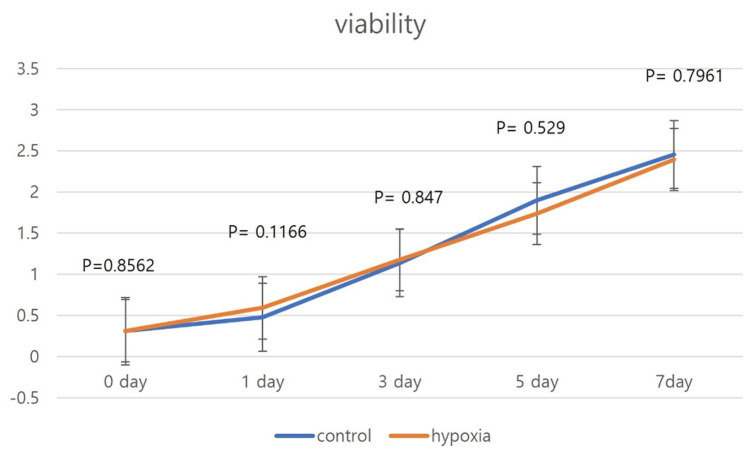
Comparison of hNTSC proliferation between hypoxic and normoxic cultivation (sample, *n* = 5). The cellular proliferation assay was performed over a 7-day period. hNTSCs from normoxic cultivation exhibited rapid proliferation after day 1 through day 7. The proliferation patterns were similar to those of MSCs grown under hypoxic cultivation. Error bars are standard errors. *t*-test was used for statistical analysis.

**Figure 5 cells-12-02360-f005:**
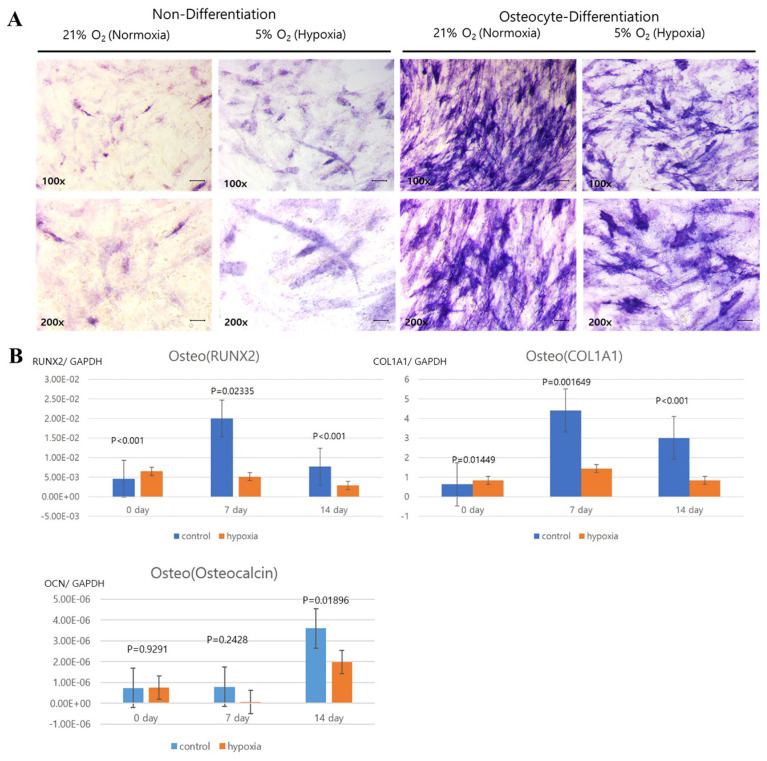
Comparison of the osteogenic differentiation potential of hNTSCs cultivated under hypoxic and normoxic conditions (sample, *n* = 5). hNTSCs cultured in osteogenic induction medium showed similar levels of alkaline phosphatase staining by visual assessment (200× magnification) between hypoxic ((**A**); **right**) and normoxic ((**A**); **left**) culture conditions (Scale bar: 10 μm). However, mRNA expression of the osteogenic differentiation markers Runx2, type I collagen (early stage of osteogenic differentiation), and osteocalcin (late markers of osteogenic differentiation), detected with RT-PCR, was higher in hNTSCs under normoxic cultivation than hypoxic cultivation (**B**). Error bars are standard errors. *t*-test was used for statistical analysis.

**Figure 6 cells-12-02360-f006:**
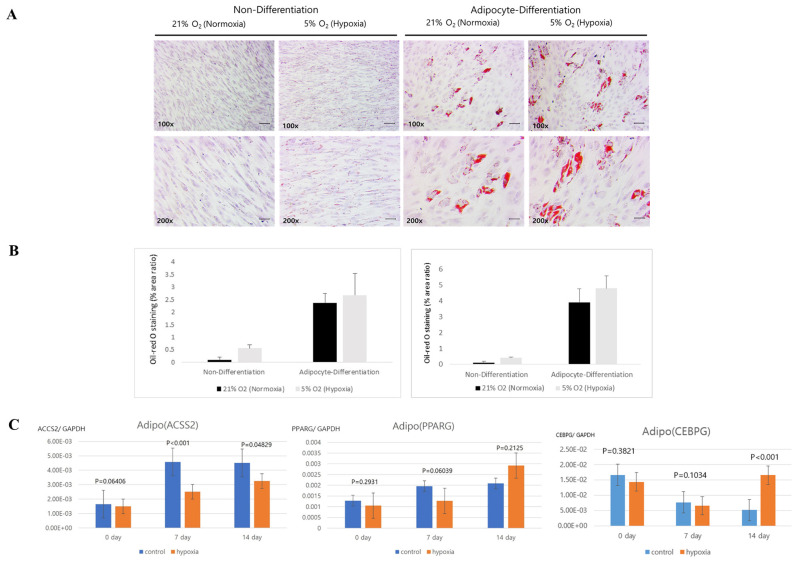
Comparison of the adipogenic differentiation potential of hNTSCs cultivated under hypoxic and normoxic conditions (sample, *n* = 5). hNTSCs cultured in adipogenic induction medium showed similar levels of intracytoplasmic lipid droplet staining with Oil Red O by visual assessment (200× magnification) between hypoxic ((**A**); **right**) and normoxic ((**A**); **left**) culture conditions (Scale bar: 10 μm). Panel show graphs the area ratio (%) of Oil Red O staining as determined by ImageJ. Error bars are standard deviations (**B**). In mRNA expression by RT-PCR, ACCS2 was significantly higher in control, but PPARγ and C/EBPα were significantly higher on day 14 of hypoxic condition. Error bars are standard errors (**C**). *t*-test was used for statistical analysis.

**Figure 7 cells-12-02360-f007:**
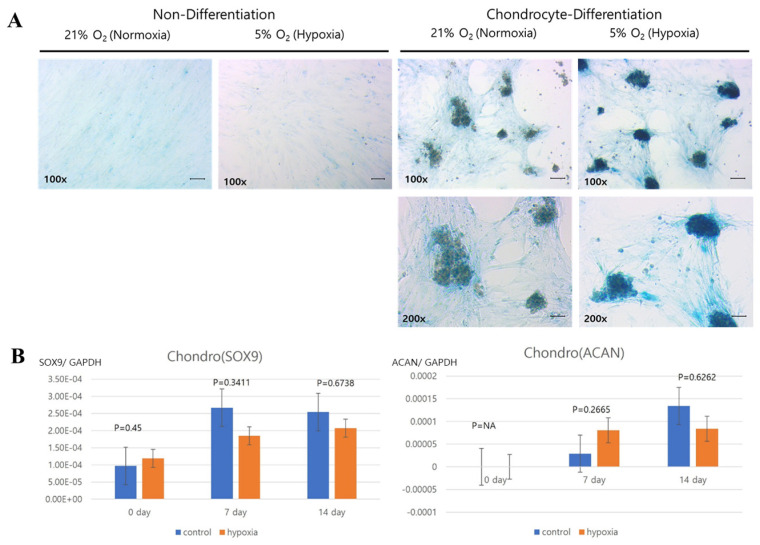
Comparison of the chondrogenic differentiation potential of hNTSCs cultivated under hypoxic and normoxic conditions (sample, *n* = 5). hNTSCs cultured in chondrogenic induction medium showed similar levels of Alcian blue staining by visual assessment (200× magnification) between hypoxic ((**A**); **right**) and normoxic ((**A**); **left**) culture conditions (Scale bar: 10 μm). hNTSCs under both culture conditions exhibited a similar pattern of increasing mRNA expression of the chondrogenic differentiation markers SOX9 and aggrecan over culture time based on RT-PCR analysis (**B**). Error bars are standard errors. *t*-test was used for statistical analysis.

**Table 1 cells-12-02360-t001:** Gene expression assays used for real-time polymerase chain reaction for multilineage differentiation.

Gene	Abbreviation	Reference Sequence	Assay Number
Type I collagen	COL1A1	NM_000088	Hs00164004_m1
Osteocalcin	OC	NM_199173	Hs01587814_g1
Runt-related transcription factor 2	Runx2	NM_001015051	Hs00231692_m1
Aggrecan	ACAN	NM_001135	Hs00153936_m1
Peroxisome proliferator-activated receptor gamma	PPARγ	NM_138712	Hs01115513_m1
Acyl-CoA synthetase short-chain family member 2	ACS	NM_001076552	Hs00218766_m1
Glyceraldehyde 3-phosphate	GAPDH	NM_002046	Hs99999905_m1

## Data Availability

No new data were created or analyzed in this study. Data sharing is not applicable to this article.

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
