# Peer review of "Characteristics of Human Nasal Turbinate Stem Cells under Hypoxic Conditions"

_cells, 2023, doi:10.3390/cells12192360_

Round 1

Reviewer 1 Report

The present research article “Characteristics of Human Nasal Turbinate Stem Cells Under Hypoxic Conditions” by Kim Do Hyun et al., investigates how hypoxia affects the human nasal inferior turbinate-derived stem cells (hNTSCs) characteristics. Authors reported that hypoxia cultural condition not modified hNTSC  proliferation, citokynes release, and multipotency. Authors performed in vitro studies to characterize the primary hNTSCs and measure the cytokines and the multilineage differentiation potential markers under normal and hypoxic conditions. This work is interesting and however, there are some concerns-in particular about as follows:

Is not clear the procedure for the hypoxic condition and should be better described: in the 2.1. Donors and cell isolation authors firstly describe in text and in figure legend 1 that hNTSCs folowing the third passage were placed in a plastic bag with the deoxidizing agent and then speak about the tissue samples in plastic bag with the deoxidizing agent.

There appears to be a significant difference regarding the number of cells seeded for the cytokine test and the proliferation assay, are the data provided, correct?

Did authors calculate the doubling time concerning the 5 lines used? is  cellular proliferation capability similar? The cytokines and proliferation assay are done for the five lines? the results are the mean?

Authors shoud be explain if  Histology and flow cytometric comparison has been done for each donor cell line. 

All figures should be increased in image quality and authors should modify ALL graphs by adding axis title and units. The statistical analysis for Figures is unclear and should be indicated if it is significantly for parameters between the normal and hypoxia groups. Furthermore, experimental sample numbers (n) should be indicated in the figure legends and type of statistical  test used.

Discussion should be improved by describing results in depth and connected way.

Author Response

Is not clear the procedure for the hypoxic condition and should be better described: in the 2.1. Donors and cell isolation authors firstly describe in text and in figure legend 1 that hNTSCs folowing the third passage were placed in a plastic bag with the deoxidizing agent and then speak about the tissue samples in plastic bag with the deoxidizing agent.

â—Ž Reply:

Cells obtained from patient tissue were primary cultured under 21% O2 conditions, and passage 3 cells were placed in a plastic bag. We changed “the tissue samples” to “the cells” to avoid confusion (Line 124).

There appears to be a significant difference regarding the number of cells seeded for the cytokine test and the proliferation assay, are the data provided, correct?

â—Ž Reply:

The cytokines assays were conducted in a 24 well plate, and the proliferation assay was conducted in a 96 well plate, and there is no error in the number of cells described.

However, as the reviewer pointed out, it seemed necessary to unify the notation of the paper, so it was changed to the format used for cytokines assays (Changed the notation from “10,000 cells” to “1 × 104 cells; Line 152).

Did authors calculate the doubling time concerning the 5 lines used? is  cellular proliferation capability similar? The cytokines and proliferation assay are done for the five lines? the results are the mean?

â—Ž Reply:

Cellular proliferation capability showed similar patterns in the cell lines evaluated. All five cell lines were analyzed and the average value was used. We added this to the methods section (Line 142 to 143, 152 to 153, and 161 to 162).

Authors shoud be explain if  Histology and flow cytometric comparison has been done for each donor cell line.

â—Ž Reply:

Histology was analyzed for each cell line, and flow cytometry analyzed one cell line. We added this to the methods section (Line 166 and 185).

All figures should be increased in image quality and authors should modify ALL graphs by adding axis title and units. The statistical analysis for Figures is unclear and should be indicated if it is significantly for parameters between the normal and hypoxia groups. Furthermore, experimental sample numbers (n) should be indicated in the figure legends and type of statistical test used.

â—Ž Reply:

Image resolution meets the standards of the cells journal. However, according to the submission guideline, images should be inserted into a manuscript word file. Therefore, the resolution may be displayed at low resolution. We have attached the original images as supplements.

We added axis titles to the graphs (Figure 4 has no units because it is a ratio).

We added sample number and statistical analysis method to the corresponding Figure legends.

Discussion should be improved by describing results in depth and connected way.

â—Ž Reply:

According to the reviewer’s comment, we summarized the results in the discussion section and added paragraphs interpreting the results (Line 333 to 374).

Reviewer 2 Report

Generally, in this article author described step-by-step procedures of novel method and well-designed protocol to establish the hypoxia culturing conditions by lowing oxygen tension with deoxidizing agent. Especially, those practical details of hypoxic cultivation set up have been illustrated very clearly by text and pictures. Author also introduced the details of methods to harvest hNTSCs from human inferior turbinate tissue. All those procedures are described very clearly in details, which make it feasible for readers to reproduce the hypoxia cultivation model for their own research. By showing those data on profiling surface markers, proliferation, as well as osteogenic, adipogenic, and chondrogenic differentiation of hNTSCs under normoxic and hypoxic conditions, author suggests hypoxic condition has no significant impact on MSC characteristics and multipotentiality of hNTSC. This work is interesting and potentially beneficial for future research on developing hNTSC based treatment or therapeutics of regenerative medicine. 

However, those data author presented on determining differentiation potential of hNTSC under normoxic and hypoxic culture conditions can’t comprehensively support author’s core conclusion of this article. The scientific quality of this article can be further improved by supplementing more data in this part. Specifically, this article needs to be revised on following points: 

Major points: 

Figure 3, 4, 5B, 6B, 7B: the metrics or units of y axis needs to be clarified, explained and clearly labelled.  

Line 297: Figure 5 A (osteocyte differentiation) shows the cell density (cells in picture) in cell culture of hypoxia look much less than in cell culture of normoxia. Because the cell density is a critical factor affecting both MSC differentiation and the intensity of alkaline phosphate staining, author needs to verify whether the cell density has been altered by hypoxia condition during osteogenic differentiation process. If yes, please discuss the possible reasons (such as alteration of cell apoptosis or proliferation) and their impact on osteogenic differentiation. 

Section 3.4: 

·      (Figure 5 A) The alkaline phosphate (ALP) is a good marker for early-stage differentiation of osteoblast, but it’s not ideal evidence to comprehensively prove the calcium deposits and osteocyte differentiation. Author needs provide additional data which give direct evidence to prove the deposit of calcium, such as Alizarin Red or von Kossa staining. 

·      (Figure 5 A) The ALP is normally recognized as markers for early stage of osteogenic differentiation, author needs to check those markers of osteoblast (bone producing) and osteocyte (stable) stage, such as osteopontin and osteocalcin. 

·      The data of Figure 6 is insufficient to comprehensively support author’s conclusion for this section. In Figure A (specifically the adipocyte differentiation panel), the positive staining of Oil Red need to be quantified (such as by using Image processing software). The gene expression levels of other adipocyte markers need to be checked.

·      The conclusion of this section can be further enhanced by supplementing data which can verify protein abundance (such as by Western blot or IF staining) of those specific markers of osteogenic, adipogenic, and chondrogenic differentiation in differentiated cells under hypoxic and normoxic conditions. 

·      The conclusion made in this section “These results suggest that hNTSCs cultured under hypoxic conditions have lower osteogenic and adipogenic differentiation potentials” (Line 292-293) and the discussion “we were able to ascertain that exposure to hypoxic conditions held no influence over the proliferation, multipotentiality (encompassing osteogenic, adipogenic, and chondrogenic differentiation),” (Line 366-368) are contradictory statements, author need to explain that. 

Minor points: 

1.    Author needs to correct some minor typographical errors: 

Line 105: 1-mm3 (1-? mm3)

Line 140 and 182: 2x104 cells (2x10cells)

Line 165: 1x105 cells (1x105 cells)

Line 315: SOX (SOX9)

Figure 5A,6A,7A Normaxia (Normoxia)

2.    Line 97: author needs to specify the gender and age range (such as younger or older age) of those patients. 

3.    Figure 2: All pictures (A) and words/numbers in charts (B) are very small and very blurry to read.

4.    Figure 2 (B): It’s better to also present summarized flow cytometry data with bar charts.  

5.    Line 140: the cell density needs to be clarified (2x10cells in one well of 24-well plates?)

6. Figure 2A, 5A ,6A 7A needs scale bar in pictures; error bars need to be defined. 

This article is written well in English and the structure of article is properly organized, which makes it easy to read and understand. The writing style can be further improved by language editing to improve syntax, such as replacing long sentences (e.g. the long sentence in line 48-51) with a few short and very well-organized sentences.

Author Response

Major points:

Figure 3, 4, 5B, 6B, 7B: the metrics or units of y axis needs to be clarified, explained and clearly labelled.

â—Ž Reply:

We added axis titles to the graphs (Figure 4 has no units because it is a ratio).

Line 297: Figure 5 A (osteocyte differentiation) shows the cell density (cells in picture) in cell culture of hypoxia look much less than in cell culture of normoxia. Because the cell density is a critical factor affecting both MSC differentiation and the intensity of alkaline phosphate staining, author needs to verify whether the cell density has been altered by hypoxia condition during osteogenic differentiation process. If yes, please discuss the possible reasons (such as alteration of cell apoptosis or proliferation) and their impact on osteogenic differentiation.

â—Ž Reply:

In this study, a duration of 72 hours of exposure to hypoxia, as evaluated through a proliferation assay, did not yield any significant reduction in cellular proliferation. On the other hand, prolonged exposure over a period of two weeks to hypoxic conditions, as assessed through a differentiation assay, resulted in a notable decline in cell density. The precise underlying mechanisms governing the demise of MSCs under conditions of oxygen deprivation remain unknown. Nonetheless, it is noteworthy that a prior study reported analogous outcomes, indicating that MSCs viability did not appear to be compromised during brief periods of hypoxic exposure (less than 72 hours); however, exposure extending to 120 hours led to an escalation in cell mortality rates (Bone. 2007;40:1078–1087). Grayson et al. documented that prolonged cultivation of MSCs under hypoxic circumstances resulted in diminished cellular proliferation, although it did not result in a concomitant elevation in apoptosis rates following 9, 16, or 24 days of culture. Hypoxia, as an explanation, exhibited inhibitory effects on their progression through the G1 phase of the cell cycle. Remarkably, despite the extended lag phase, MSCs exhibited limited variability in their expression of p21 and p53 at the initial stages of hypoxic culture, indicative of the fact that hypoxia did not induce the expression of these proteins associated with the regulation of apoptosis in MSCs (J Cell Physiol 2006;207:331–9). Consequently, these findings, when taken in conjunction with our own observations, postulate that hypoxia primarily induces a modest degree of cell death, while the surviving MSCs population remains capable of proliferative activity.

Additionally, it has been documented in the literature that low oxygen tension levels exert suppressive effects on the osteo-plasticity of MSCs (Aging Cell. 2007;6:745–757; Bone. 2007;40:1078–1087). These effects manifest as an absence of osteogenic differentiation under conditions of 3% oxygen (Aging Cell. 2007;6:745–757). Furthermore, exposure of MSCs to hypoxia has been observed to result in a persistent downregulation, extending up to 14 days post-exposure, of critical markers such as cbfa-1/Runx2, osteocalcin, and type I collagen (Bone. 2007;40:1078–1087). These collective findings provide a plausible explanation for the observed reduction in osteogenic differentiation capacity in MSCs consequent to prolonged exposure to hypoxic conditions.

We have added the above to the discussion section (Line 341 to 367).

Section 3.4:

  • (Figure 5 A) The alkaline phosphate (ALP) is a good marker for early-stage differentiation of osteoblast, but it’s not ideal evidence to comprehensively prove the calcium deposits and osteocyte differentiation. Author needs provide additional data which give direct evidence to prove the deposit of calcium, such as Alizarin Red or von Kossa staining.

â—Ž Reply:

Staining takes about 3 weeks, and we cannot meet the revision due day (10 days).

Instead, PCR results of osteocalcin (late markers of osteogenic differentiation) were added (Figure 5B).

(Figure 5 A) The ALP is normally recognized as markers for early stage of osteogenic differentiation, author needs to check those markers of osteoblast (bone producing) and osteocyte (stable) stage, such as osteopontin and osteocalcin.

â—Ž Reply:

We added PCR results for osteocalcin (late markers of osteogenic differentiation) (Figure 5B).

  • The data of Figure 6 is insufficient to comprehensively support author’s conclusion for this section. In Figure A (specifically the adipocyte differentiation panel), the positive staining of Oil Red need to be quantified (such as by using Image processing software). The gene expression levels of other adipocyte markers need to be checked.

â—Ž Reply:

We added the results of Oil-red O staining image analysis using the ImageJ program (Laboratory for Optical and Computational Instrumentation, WI, USA) (Figure 6B). Also, C/EBPα marker results analyzed by PCR were also added (Figure 6C).

As a result of Oil-red O staining image analysis, it was confirmed that adipogenic differentiation was better under hypoxic conditions. Also, in mRNA expression by RT-PCR, ACCS2 was significantly higher in control, but PPARγ and C/EBPα were significantly higher on day 14 of hypoxic condition.

C/EBPδ is a critical determinant for the early stage of adipocyte differentiation. C/EBPδ are expressed in the early stage of adipogenesis in culture and induce PPARγ and C/EBPα expression (Mol Cell Biol. 2019 May 14;39(11):e00601-18). Therefore, it is possible to explain that although the expression levels of C/EBPα and PPARγ were lower than control on day 7, they both significantly increased compared to control on day 14. However, considering that PPARγ and C/EBPα are critical transcription factors in adipogenesis (Genes Dev. 2002 Jan 1; 16(1): 22–26), based on the results of PPARγ and C/EBPα, it can be judged hypoxic conditions promote adipogenic differentiation compared to the control group (Line 367 to 374).

We have revised the Figure 6 legend with the addition of results (Line 319 to 323).

  • The conclusion of this section can be further enhanced by supplementing data which can verify protein abundance (such as by Western blot or IF staining) of those specific markers of osteogenic, adipogenic, and chondrogenic differentiation in differentiated cells under hypoxic and normoxic conditions.

â—Ž Reply:

As the reviewer comments, supplementing data may provide additional information about differentiation. However, there were limitations to the items evaluated under limited resources and time. Therefore, we have added to the text as a limitation of the study a mention of validation by additional tests such as Western blot or immunofluorescence screening (Line 433 to 434).

  • The conclusion made in this section “These results suggest that hNTSCs cultured under hypoxic conditions have lower osteogenic and adipogenic differentiation potentials” (Line 292-293) and the discussion “we were able to ascertain that exposure to hypoxic conditions held no influence over the proliferation, multipotentiality (encompassing osteogenic, adipogenic, and chondrogenic differentiation),” (Line 366-368) are contradictory statements, author need to explain that.

â—Ž Reply:

Hypoxia did not exert any discernible impact on the proliferation of hNTSCs. Furthermore, the cytokine expression patterns exhibited noteworthy similarities between the hypoxic and normoxic cohorts throughout the duration of cultivation. Nonetheless, it is noteworthy that hypoxic conditions did suppress the osteogenic and promote adipogenic differentiation potential of hNTSCs, while chondrogenic differentiation remained relatively unaffected. There was no statistical difference in cartilage differentiation, but it tended to be relatively low when quantitatively analyzed by PCR.

There was an error in the way we phrased it and lines 366-368 have been rewritten to resolve any confusion.

Minor points:

  1. Author needs to correct some minor typographical errors:

Line 105: 1-mm3 (1-? mm3)

Line 140 and 182: 2x104 cells (2x104 cells)

Line 165: 1x105 cells (1x105 cells)

Line 315: SOX (SOX9)

Figure 5A,6A,7A Normaxia (Normoxia)

â—Ž Reply:

We corrected typos.

  1. Line 97: author needs to specify the gender and age range (such as younger or older age) of those patients.

â—Ž Reply:

The five donors, three of whom were male and two of whom were male, were aged between 24 and 47 years old (Line 101 to 102).

  1. Figure 2: All pictures (A) and words/numbers in charts (B) are very small and very blurry to read.

â—Ž Reply:

We rechecked and replaced the images in Figure 2. However, according to the submission guideline, images should be inserted into a manuscript word file. Therefore, Figures shown in manuscript Word file may be displayed in low resolution. We have attached the original images as supplements.

**** 4. Figure 2 (B): It’s better to also present summarized flow cytometry data with bar charts. 

â—Ž Reply:

Flow cytometry was performed with only one cell line (clarified in this revision).

  1. Line 140: the cell density needs to be clarified (2x104 cells in one well of 24-well plates?)

â—Ž Reply:

We fixed the typo.

**** 6. Figure 2A, 5A ,6A 7A needs scale bar in pictures; error bars need to be defined.

â—Ž Reply:

Scale bar is inserted (Figure 2A, 5A ,6A 7A).

Error bars ​​are standard errors. We revised figure legends.

Comments on the Quality of English Language

This article is written well in English and the structure of article is properly organized, which makes it easy to read and understand. The writing style can be further improved by language editing to improve syntax, such as replacing long sentences (e.g. the long sentence in line 48-51) with a few short and very well-organized sentences.

â—Ž Reply:

We divided Lines 48-51 as follows to improve readability:

These investigations center around the enhancement of factors like media composition, oxygen tension, and substrate coatings. The aim is to bolster the expansion of MSCs, their potential to differentiate, and their broader effectiveness in the therapeutic realm of regenerative medicine.

We added a paragraph summarizing the research results at the beginning of the discussion section as follows:

Our study revealed that hNTSCs subjected to hypoxic conditions maintained their distinctive MSC phenotype. This assertion was substantiated through FACS, which confirmed the absence of hematopoietic markers and the presence of MSC markers. Hypoxia did not exert any discernible impact on the proliferation of hNTSCs. Furthermore, the cytokine expression patterns exhibited noteworthy similarities between the hypoxic and normoxic cohorts throughout the duration of cultivation. Nonetheless, it is noteworthy that hypoxic conditions did suppress the osteogenic and promote adipogenic differentiation potential of hNTSCs, while chondrogenic differentiation remained relatively unaffected.

Round 2

Reviewer 1 Report

the revised version is improved